# Geometric Nonlinear Model for Prediction of Frequency–Temperature Behavior of SAW Devices for Nanosensor Applications

**DOI:** 10.3390/s20154237

**Published:** 2020-07-29

**Authors:** Zhenglin Chen, Qiaozhen Zhang, Congcong Li, Sulei Fu, Xiaojun Qiu, Xiaoyu Wang, Haodong Wu

**Affiliations:** 1School of Electronic Science and Engineering, Nanjing University, Nanjing 210093, China; zlchen1988@yeah.net (Z.C.); xjqiu@nju.edu.cn (X.Q.); 2College of Information, Mechanical and Electrical Engineering, Shanghai Normal University, Shanghai 200234, China; 3Key Laboratory of Modern Acoustics, Ministry of Education, Department of Acoustic Science and Engineering, School of Physics, Nanjing University, Nanjing 210093, China; mf1822006@smail.nju.edu.cn (C.L.); wangxy512@163.com (X.W.); 4Key Laboratory of Advanced Materials (MOE), School of Materials Science and Engineering, Tsinghua University, Beijing 100084, China; fusulei@mail.tsinghua.edu.cn

**Keywords:** Lagrangian equations, three-dimensional periodic SAW structure, geometric nonlinearity, thermal expansion, the frequency–temperature characteristics

## Abstract

Surface acoustic wave (SAW)-based sensors have become highly valued for their use as nanosensors in industrial applications. Accurate prediction of the thermal stability is a key problem for sensor design. In this work, a numerical tool based on the finite element method combined with piezoelectric Lagrangian equations has been developed to accurately predict the thermal sensitivity characteristics of surface acoustic wave devices. Theoretical analysis for the geometric nonlinearity contributing to the frequency–temperature characteristic and material constants’ dependency on temperature were taken into consideration. The thermomechanical equilibrium equation built on the three-dimensional finite element method (3D-FEM) mesh node took mesh movement into account because thermal expansion was employed. The frequency–temperature characteristics of different SAW modes, including Rayleigh waves and leaky waves excited on a piezoelectric substrate of quartz or lithium tantalate, respectively, were calculated. The theoretical accuracy of the proposed numerical tool was verified by experiments.

## 1. Introduction

The miniaturization of sensors has attracted much attention in several important applications, especially in industries such as biological and medical fields [1], radio frequency (RF) devices for 5G [2,3,4], and space exploration [5]. Surface acoustic wave (SAW)-based sensors, including temperature sensors and non-temperature sensors, are expected to achieve a nanoscale structure with the rapid development in micro/nanoscience and semiconductor technology [6,7,8,9], and are widely used for humidity [10,11], temperature [12,13], and pressure [14] sensing applications. Moreover, traditional sensing technology mostly faces problems in terms of wired installation and power supply requirements. SAW-based sensors have the advantages of being wireless and battery-free, and do not need a separate power supply, which makes wireless installation at particularly inaccessible locations possible [15]. Since the photolithography technique allows for a submicron-scale line width [16], miniaturized SAW devices can be readily fabricated. Therefore, SAW-based sensors have become highly valued as nanosensors in industrial applications. Nevertheless, the frequency–temperature characteristic of SAW-based sensors calls for intensive study. For sensor design, accurate prediction of the thermal stability is a key problem. However, accurate prediction of the thermal behavior of SAW-based sensors is difficult because previous theoretical work has made a series of hypotheses related to homogeneous physical fields along the aperture direction, geometric linearity, rectangle geometrical shapes of electrodes, and so on.

The frequency–temperature characteristic of surface acoustic waves excited on piezoelectric substrates has been studied using different methods [17,18,19,20]. During the early stage of research, studies were mainly concerned with the dependency of crystal cuts and propagation directions on temperature in a SAW device [21,22]. Campbell and Jones [23] proposed a general approach in which the electrode mass load is taken into account and the material constant of the electrode and piezoelectric substrate is used as a function of temperature. Yong et al. [18,24,25] studied the frequency–temperature behavior of quartz by means of building piezoelectric Lagrangian equations performed with the two-dimensional periodic finite element method (2D-FEM) model and three-dimensional FEM model (3D-FEM) of a SAW structure. Finite element analysis (FEA) and a boundary integral method (BIM) were developed by Pastureaud et al. [20], where Green’s function and harmonic admittance as a function of the temperature and an infinite and periodic structure were assumed. Finite element analysis/the boundary element method (FEA/BEM) combined with material coefficient perturbation has been investigated by Garcia et al. [26] based on a two-dimensional finite element analysis of periodic structures. Wang et al. [27] extended the FEM/BEM method to include the electrode Lame constant with temperature dependency, except for the mass load and the numerical result was in good agreement with experimental results. In our work [28], by considering the material parameters and thermal expansion, FEM commercial software COMSOL Multiphysics was utilized to analyze the frequency–temperature response of SAW temperature sensors. The methods reported in the above previous works offer solutions for prediction of frequency–temperature characteristic, but they ignored the nonlinear effect owing to temperature-induced thermal stress and strain tensors. Lately, a remarkable solution of a weak form of nonlinear FEM model for calculating the thermal sensitivity on arbitrary layered structures has been reported [29], which demonstrated that the nonlinear effect caused by thermal stress and strain tensors between the substrate and electrodes is not negligible.

Therefore, this paper aims to develop a geometric nonlinearity FEM model (GN-FEM) for accurately predicting the frequency–temperature characteristic of SAW-based sensors. First, a quasi-3D periodic SAW structure is analyzed on the basis of piezoelectric Lagrangian equations. The proposed GN-FEM is employed to establish the thermomechanical equilibrium equation for the moving mesh node of the FEM model. Meanwhile, the material constants, including elastic constant, coupling constant, dielectric permittivity constant, and density, as well as the thermal stress and strain of the substrate and electrode as a function of temperature, are considered. Furthermore, the trapezoid shape of the electrode combined with practical processing technology is taken into account. Finally, the relative frequency shift of SAW devices with different piezoelectric materials is investigated to verify the model. The calculated frequency–temperature characteristics are in good agreement with the experiment results. Meanwhile, the dependencies of the relative frequency shift, turnover temperature, and temperature coefficient of frequency (TCF) value on the metallization ratio and electrode thickness are analyzed, which provide the solution for the design of SAW-based sensors.

The remainder of this paper is divided into three sections. The second section is devoted to a theoretical analysis of geometric nonlinear models for the frequency–temperature behavior of SAW devices based on Lagrangian equations. In the third section, the relative frequency shift, the turnover temperature, and the TCF value obtained from the theoretical calculation of different mode waves are predicted and compared with the experimental results. Finally, conclusions are discussed in detail in the last section of the paper. 

## 2. Theoretical Analysis of the Frequency–Temperature Behavior Characteristic

### 2.1. Thermal Sensitivity Equations of the Piezoelectric Substrate

The piezoelectric constitutive relation of displacement and electric field can be expressed as: (1)$${Τ}_{ij}=c_{ijkl}S_{kl}-e_{kij}E_{k}$$
(2)$$D_{i}=e_{kij}S_{kl}+\varepsilon_{ik}E_{k}$$
where ${Τ}_{ij}$ and $S_{kl}$ are the stress and strain tensors, respectively; $c_{ijkl}$, $e_{kij}$, and $\varepsilon_{ik}$ are the stiffness constant, piezoelectric stress constant, and dielectric permittivity constant, respectively; and $D_{i}$ and $E_{k}$ are the electric displacement vector and electric field, respectively.

According to the electrostatics, the relations between the electric displacement $D_{i}$, electric field $E_{k}$, electric potential $\phi_{k}$, and charge density $\rho_{s}$ are defined by:(3)D=εE
(4)$$\nabla \cdot D=\rho_{s}$$
(5)E=−∇ϕ
where ∇ is the Laplace operator.

Because of thermal expansion or contraction at a given temperature, deformation of the SAW structure occurs, leading to the mesh node of the FEM model moving in a certain direction. The geometric nonlinear relation between the thermal strain tensors and partial displacement can be expressed as [30,31]:(6)$$\varepsilon_{x}=\frac{\partial u}{\partial x}+\frac{1}{2}\left[{\left(\frac{\partial u}{\partial x}\right)}^{2}+{\left(\frac{\partial v}{\partial x}\right)}^{2}+{\left(\frac{\partial w}{\partial x}\right)}^{2}\right]$$
(7)$$\varepsilon_{y}=\frac{\partial v}{\partial y}+\frac{1}{2}\left[{\left(\frac{\partial u}{\partial y}\right)}^{2}+{\left(\frac{\partial v}{\partial y}\right)}^{2}+{\left(\frac{\partial w}{\partial y}\right)}^{2}\right]$$
(8)$$\varepsilon_{z}=\frac{\partial w}{\partial z}+\frac{1}{2}\left[{\left(\frac{\partial u}{\partial z}\right)}^{2}+{\left(\frac{\partial v}{\partial z}\right)}^{2}+{\left(\frac{\partial w}{\partial z}\right)}^{2}\right]$$
(9)$$\gamma_{xy}=\frac{\partial v}{\partial x}+\frac{\partial u}{\partial y}+\frac{\partial u}{\partial x}\frac{\partial u}{\partial y}+\frac{\partial v}{\partial x}\frac{\partial v}{\partial y}+\frac{\partial w}{\partial x}\frac{\partial w}{\partial y}$$
(10)$$\gamma_{xz}=\frac{\partial w}{\partial x}+\frac{\partial u}{\partial z}+\frac{\partial u}{\partial x}\frac{\partial u}{\partial z}+\frac{\partial v}{\partial x}\frac{\partial v}{\partial z}+\frac{\partial w}{\partial x}\frac{\partial w}{\partial z}$$
(11)$$\gamma_{yz}=\frac{\partial v}{\partial x}+\frac{\partial w}{\partial y}+\frac{\partial u}{\partial z}\frac{\partial u}{\partial y}+\frac{\partial v}{\partial z}\frac{\partial v}{\partial y}+\frac{\partial w}{\partial z}\frac{\partial w}{\partial y}$$

The relation of strain and displacement with the combination of the thermal expansion coefficient is given by:(12)$$S_{ij}=\frac{1}{2}\left(\alpha_{kj}u_{k,i}+\alpha_{ki}u_{k,j}\right),$$
where α represents the thermal expansion coefficients.

On the other hand, the material parameters, including the elastic constant, coupling constant, dielectric permittivity constant, and so on, as a function of temperature, can be written as [24,27]:(13)$$c_{ijkl}^{\theta }=c_{ijkl}+c_{ijkl}^{\left(1\right)}\theta +c_{ijkl}^{\left(2\right)}\theta^{2}+c_{ijkl}^{\left(3\right)}\theta^{3}$$
(14)$$e_{ijk}^{\theta }=e_{ijk}+e_{ijk}^{\left(1\right)}\theta +e_{ijk}^{\left(2\right)}\theta^{2}+e_{ijk}^{\left(3\right)}\theta^{3}$$
(15)$$\varepsilon_{ik}^{\theta }=\varepsilon_{ik}+\varepsilon_{ik}^{\left(1\right)}\theta +\varepsilon_{ik}^{\left(2\right)}\theta^{2}+\varepsilon_{ik}^{\left(3\right)}\theta^{3}$$
(16)$$\alpha_{ik}^{\theta }=\delta_{ik}+\alpha_{ik}^{\left(1\right)}\theta +\alpha_{ik}^{\left(2\right)}\theta^{2}+\alpha_{ik}^{\left(3\right)}\theta^{3}$$
(17)$$\rho^{\theta }=\rho +\rho^{\left(1\right)}\theta +\rho^{\left(2\right)}\theta^{2}+\rho^{\left(3\right)}\theta^{3}$$
(18)$$r^{\theta }=r+r^{\left(1\right)}\theta +r^{\left(2\right)}\theta^{2}+r^{\left(3\right)}\theta^{3}$$
(19)$$g^{\theta }=g+g^{\left(1\right)}\theta +g^{\left(2\right)}\theta^{2}+g^{\left(3\right)}\theta^{3}$$
(20)$$\theta =\left(T-T_{0}\right)$$
where $c_{ijkl}^{\left(n\right)}$ is the nth-order temperature coefficient of elastic constants, $e_{ijk}^{\left(n\right)}$ is the nth-order temperature coefficient of piezoelectric coupling constants, and $\varepsilon_{ik}^{\left(n\right)}$ is the nth-order temperature coefficient of dielectric permittivity constants. $\delta_{ik}$ is the Kronecker delta and $\alpha_{ik}^{\left(n\right)}$ represents the nth-order temperature coefficients of thermal expansion. $\rho^{\left(n\right)}$ denotes the nth-order temperature coefficient of the density. $r^{\left(n\right)}$ and $g^{\left(n\right)}$ denote the nth-order temperature coefficient of the first and second Lame constant of electrode metal, respectively. $T_{0}$ is the reference temperature ($T_{0}=25\;{}^{\circ}C$) and θ is the temperature difference.

The motion Equation (1) and electrostatics Equation (2) must satisfy Newton’s equations and Maxwell’s equations, respectively. If there is no external force applied, the equilibrium equation of the piezoelectric relation can be described as below:(21)$$\nabla^{\prime }T=\rho \ddot{u}$$
(22)$$\nabla_{d}D=0$$

### 2.2. Quasi-Three-Dimensional FEM Modeling

The Quasi-3D FEM method is an effective tool for analyzing piezoelectric devices. Figure 1a shows an illustration of the finite-length three-dimensional structure of SAW devices. In order to simplify the solution and achieve a calculation of the thermal sensitivity of SAW devices under the condition of ensuring a good enough accuracy, the finite-length 3D-FEM model was decomposed into a single finger structure with half period interdigital transducer (IDT) p (2×p=λ, where λ is the SAW wavelength), electrode width *a,* and thickness *h*, as shown in Figure 1b. Besides, we assumed that the periodic boundary condition was set to the side of the model for extending it in the X-direction to infinity. On the other hand, a substrate with a perfect matching layer set to the bottom for absorbing the wave propagated into the substrate was constructed. Additionally, it can be seen that the geometric shape of the electrode is trapezoid for coinciding with practical processing and the mesh size of the region below the electrode is smaller than others of the substrate due to the energy of the acoustic surface wave mainly focusing on the surface of the piezoelectric medium.

According to Figure 1b, the periodic boundary condition employed on the model side of the single finger reveals that the displacement and electric potentials vector located on the left side (A) has an antisymmetric relation with the vector located on the right side (B), so the relation of the displacement and electric potentials vector can be written as: (23)$${\left[\begin{array}{c} u_{x}\\ u_{y}\\ \begin{array}{c} u_{z}\\ \phi \end{array} \end{array}\right]}_{A}=-{\left[\begin{array}{c} u_{x}\\ u_{y}\\ \begin{array}{c} u_{z}\\ \phi \end{array} \end{array}\right]}_{B.}$$

## 3. Results and Discussion

The development of SAW-based sensors on quartz substrates was partially driven by the requirement for wireless temperature measurement systems due to the availability of temperature compensated cuts and the good sensitivity to strain [32,33]. The fundamental material constants, the third-order material constants, and the temperature coefficients of quartz and aluminum were taken from the reference [24]. The thermomechanical equilibrium equation based on the geometric nonlinearity of the quasi-3D FEM model, including thermal stress and strain tensors, was solved by the finite element method. The trapezoidal angle of the electrode shape was set to about 7° according to the SAW devices fabricated and the reference temperature was set to room temperature (*T*_0_ = 25 °C).

### 3.1. Simulation of Rayleigh-Type Acoustic Surface Wave Devices Excited on Quartz

A half-wavelength FEM model of the frequency–temperature behavior of quartz with a thickness of 8 λ and perfect matched layer (PML) with a thickness of 3 λ was modeled. In this case, 2*p* was 7 μm, the normalized electrode thickness *h*/2*p* was 1.8%, and the normalized metallization ratio, namely *a*/*p*, ranged from 0.3 to 0.7. The cut angles ranged from 35.0° to 37.0° in increments of 1° and included 42.75°. Correspondingly, the finite element mesh of the periodic 3D-FEM model was made of free tetrahedral elements and the discretization was set as the quadratic serendipity element in this analysis, in which about 34,902 triangular elements, about 902 quadrilateral elements, and a degree of freedom (DOF) of about 751,146 could be obtained. For electrical boundary condition, the exciting voltage V=1×exp(jωt) is performed on the electrode, where ω is the angular frequency. Therefore, the relative frequency shift of the quartz substrate was investigated as shown in Figure 2. It is obvious that the metallization ratio has a great influence on the relative frequency shift characteristic. The turnover temperature of quartz with cut angles ranging from 35.0° to 37.0° is within the range of 0 to 40 °C, while the turnover temperature of 42.75° cut quartz is less than −40 °C. Therefore, SAW devices on quartz with a cut angle range of 35.0° to 37.0° are suitable for processing into SAW based non-temperature sensors, such as SAW filters. Meanwhile, the SAW devices on 42.75° cut quartz are suitable for processing into SAW-based temperature sensors.

According to the numerical analysis above, the relation between the turnover temperature of quartz with different cut angles and the electrode metallization ratio is illustrated in Figure 3. The results showed that the turnover temperature gradually decreases with the increase of the metallization ratio and cut angle of quartz. On the other hand, when the parameter of the cut angle and metallization ratio is set to points A, B, and C, the corresponding cut angle and metallization ratio are suitable parameters for designing non-temperature sensors such as SAW filters or resonators, because the turnover temperature is near room temperature.

Furthermore, a half-wavelength FEM model of SAW with the same parameters as listed above was analyzed, with the electrode thickness *h*/2*p* ranging from 0.01 to 0.05 λ in an operating temperature range of −40 to +100 °C. In this case, the normalized metallization ratio *a*/*p* was fixed at 0.5. The relative frequency shift of the quartz substrate as a function of temperature was drawn and is presented in Figure 4. It was found that the electrode thickness has a larger effect on the relative frequency shift than that of the metallization ratio, and the SAW devices on the quartz substrate can not only be used to fabricate SAW-based temperature sensors, but also non-temperature sensors, by means of adjusting the electrode thickness. Moreover, only when the electrode thickness *h*/2*p* increases from 0.01 to 0.04 λ does the turnover temperature of quartz with cut angles ranging from 35.0° to 37.0° exist within the range of 0 to 40 °C.

Based on the calculated relative frequency shift curve shown in Figure 4, the dependence of the turnover temperature on the electrode thicknesses is obtained. As shown in Figure 5, the turnover temperature monotonically decreases with increasing electrode thickness for cut angle of Y-X 35°~36° quartz and the values for the turnover temperature decrease with increasing cut angle. In addition, by comparing Figure 3 and Figure 5, the variation of electrode thickness has a larger effect on the frequency–temperature characteristic than that of the metallization ratio. When the parameters of the cut angle and electrode thickness are set to points A, B, and C, the best parameters for designing SAW-based non-temperature sensors with a good frequency–temperature characteristic were obtained, and the turnover temperature was near room temperature.

### 3.2. Experiment of Rayleigh-Type Acoustic Surface Wave Devices Excited on Quartz

To confirm the validity and accuracy of the proposed method based on the simulation presented above, one-port SAW resonators were fabricated. Figure 6 shows an optical image and the measured frequency response of a one-port SAW resonator based on quartz. The device parameters for this SAW resonator are given in Table 1. As shown in Figure 6b, there exists obvious resonance, and the resonant frequency is 521.437 MHz at a temperature of 25 °C, and that at 100 °C is decreased to 524.275 MHz.

For demonstration, experimental samples fabricated on a piezoelectric substrate of quartz with rotated Y-cut angles of 35°, 36°, 37°, and 42.75° were investigated. In this case, Rayleigh wave was excited. To obtain their temperature behavior, those devices were tested at an operating temperature ranging from −40 to +100 °C. The thickness of the aluminum electrode *h*/2*p* was 1.8%, the pairs of IDT were 150, and the number of grating reflectors was 35 on both sides of the IDT. The aperture width of the resonator was 32 λ and the metallization ratio of the IDT and reflector was 0.5. The relative frequency shift of the numerical and experimental result is shown in Figure 7. The relative frequency shift curves increase first to the maximum and then decrease with increasing T, except for the curve for the case of 42.75°-YX° quartz.

As shown in Figure 7a, the relative frequency shift characteristic of the 35°-YX° quartz resonator at 417 MHz was investigated. The difference of the turnover temperature between the measured results (30.2 °C) and the GN-FEM numerical results (29.9 °C) is 0.3 °C, while that of FEM/BEM is about 40.02 °C. Figure 7b shows the dependency of the relative frequency shift characteristic on the 36°-YX quartz. The difference of the turnover temperature obtained from the experimental results (19 °C) and the GN-FEM theoretical results (19.9 °C) is 0.9 °C, and that of FEM/BEM is about 30 °C. As for the 37°-YX° quartz, the relative frequency shift characteristic’s dependency on temperature is illustrated in Figure 7c. The turnover temperature of the measurement (9.7 °C) is 0.3 °C greater than that of the GN-FEM analytical results (10 °C), and that of FEM/BEM is approximately 20 °C. Lastly, the relative frequency shift curve of the 42.75°-YX° quartz resonator is plotted in Figure 7d. It can be seen that the trend of the relative frequency shift of the experimental result follows that of the proposed theoretical result, but the intersection point of the two curves pictured by the experimental result and FEM/BEM method occurred at about 47 °C. However, Figure 6 shows that the measured relative frequency shift is in good agreement with that of the proposed methods and the discrepancies between them are less than ± 1 °C. The proposed theoretical analysis for Rayleigh-type SAW-based sensors can be used to accurately predict the frequency–temperature characteristic.

### 3.3. Simulation of Leaky-Type Surface Acoustic Wave Devices Excited on Lithium Tantalate

It is known that a leaky acoustic surface wave propagates on the surface of a semi-infinite piezoelectric substrate with bulk acoustic waves radiating into the substrate, but the leaky wave excited on lithium tantalate (42° Y-X LT) or LST quartz (17° Y-X quartz) is usually utilized for most radio-frequency designs [20,27]. In this work, a half-wavelength FEM model of the thermal sensitivity used a 42° Y-X LT thin plate as the piezoelectric substrate, with a thickness of 3 λ, and PML with a thickness of 2 λ was modeled. The wavelength λ, namely 2*p*, was set as 4 μm. The electrode film was polycrystalline Al metal with a thickness *h*/2*p* of 0.08 and the electrode metallization ratio *a*/*p* was set to range from 0.4 to 0.8. The finite element mesh comprised free tetrahedral elements and the discretization was set as a quadratic serendipity element, including about 13,514 triangular elements, about 392 quadrilateral elements, and a degree of freedom (DOF) of about 166,512. Material constants with a temperature dependency and the thermal expansion were taken into consideration. The fundamental material constants and the nth-order temperature coefficients of LT were taken from Murota and Shimizu [22]. Without losing generality and the purpose for confirming the validity of the GN-FEM model, the relative frequency shift of the resonance frequency of 42° Y-X LT and its TCF value with a metallization ratio dependency was investigated as shown in Figure 8.

Moreover, the same FEM model with an electrode thickness *h*/2*p* ranging from 0.06 to 0.1 in an operating temperature range of 25 °C to + 145 °C was analyzed. The relation of the relative frequency shift with different electrode thicknesses and temperatures was drawn and is presented in Figure 9. It was found that the curve of the TCF value is not linear; that is to say, the variation of the electrode thickness has a larger effect on the relative frequency shift than that of the metallization ratio.

### 3.4. Experiment of Leaky-Type Surface Acoustic Wave Devices Excited on Lithium Tantalate

As shown in Figure 10, the experimental data taken from Pastureaud et al. [20] and the numerical result calculated by the FEA/BIM method were compared. It was found that the simulation results calculated by the GN-FEM model are more consistent with the experimental data than those of the FEA/BIM model. Furthermore, the relative frequency shift with temperature dependency is not strictly linear, because the geometric nonlinearity factor and the high-order temperature coefficients of material constants are taken into consideration in the FEM model. Nevertheless, the good agreement between the numerical analysis and experimental results confirms the efficiency of the proposed theoretical analysis for investigating the thermal sensitivity of leaky-type SAW-based sensors.

## 4. Conclusions

In conclusion, a numerical analysis was developed in this study to accurately predict the frequency–temperature characteristic of SAW-based sensors using the periodic 3D-FEM model, which includes the geometric nonlinear effect caused by thermal expansion of the electrodes and piezoelectric substrate. SAW-based sensors on quartz or LT thin plates were investigated and the relative frequency shifts as a function of the cut angle, metallization ratio, and electrode thickness were calculated. This investigation provides important guidance for SAW-based sensor design. For example, the calculation results show that the SAW devices on quartz with cut angles ranging from 35.0° to 37.0° are suitable for use as non-temperature sensors, such as SAW filters, and those on 42.75°–cut quartz are suitable for application as temperature sensors. The frequency–temperature laws of the proposed theoretical result were in good agreement with those of the experimental results. More significantly, the stiffness matrix, mass matrix, and damp matrix of the FEM model as a function of the temperature can be obtained. This means that the dependency of the admittance, phase velocity, and electromechanical coupling factor on the temperature can be further analyzed. In future work, the proposed method can be extended to accurately predict temperature behavior for full-scale SAW-based nanosensors using the latest algorithm of hierarchical cascading technology (HCT). 

## Figures and Tables

**Figure 1 sensors-20-04237-f001:**
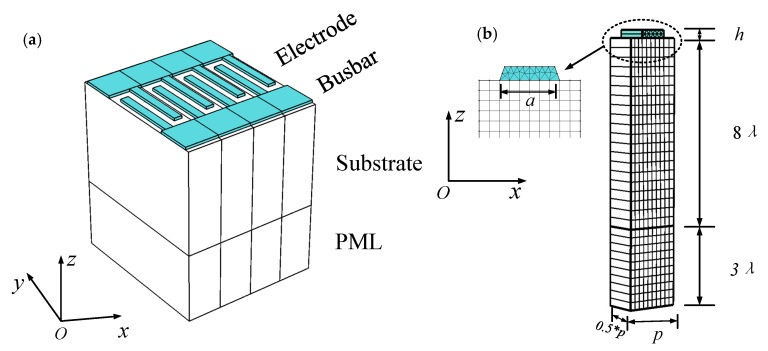
Schematic diagram of the three-dimensional model. (**a**) The geometrical structure of SAW devices; (**b**) quasi-3D FEM mesh model of SAW devices.

**Figure 2 sensors-20-04237-f002:**
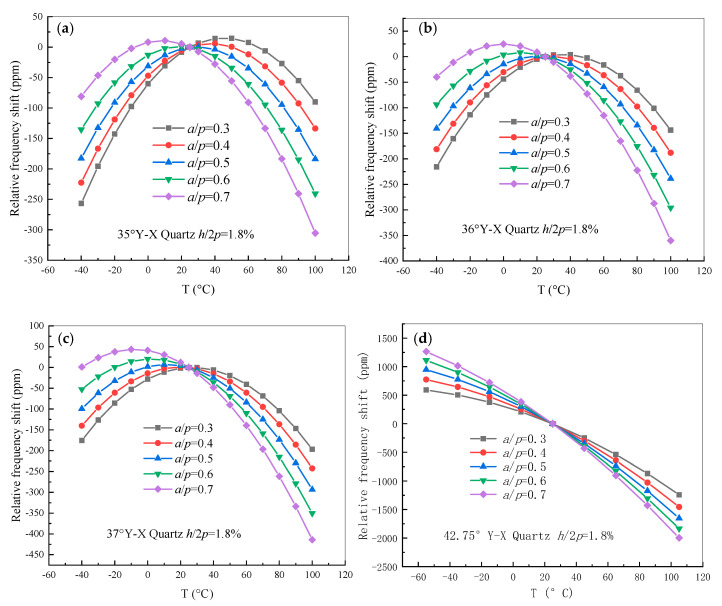
The relative frequency shift of quartz with different metallization ratios. (**a**) 35.0° Y-X quartz; (**b**) 36.0° Y-X quartz; (**c**) 37.0° Y-X quartz; (**d**) 42.75° Y-X quartz.

**Figure 3 sensors-20-04237-f003:**
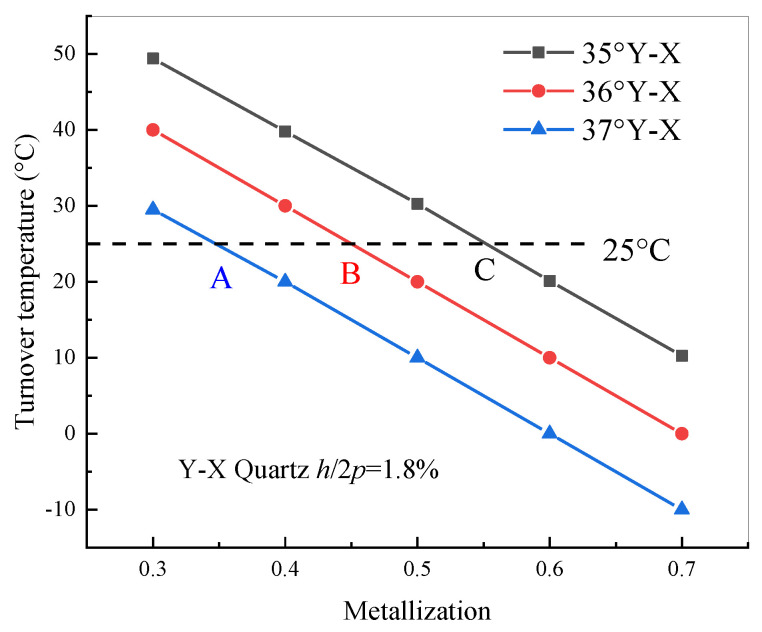
Turnover temperature of quartz with different cut angles and IDT with the metallization ratio.

**Figure 4 sensors-20-04237-f004:**
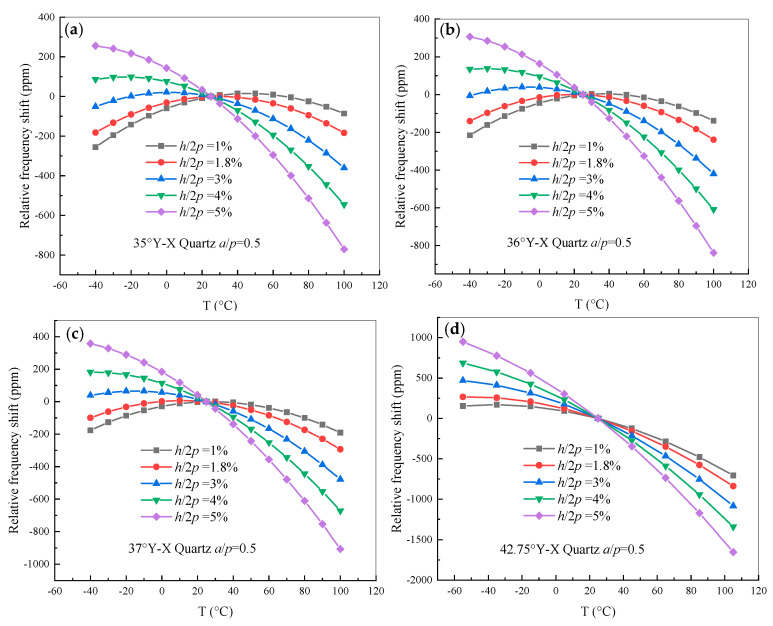
The relative frequency shift of quartz with different electrode thicknesses. (**a**) 35.0° Y-X quartz; (**b**) 36.0° Y-X quartz; (**c**) 37.0° Y-X quartz; (**d**) 42.75° Y-X quartz.

**Figure 5 sensors-20-04237-f005:**
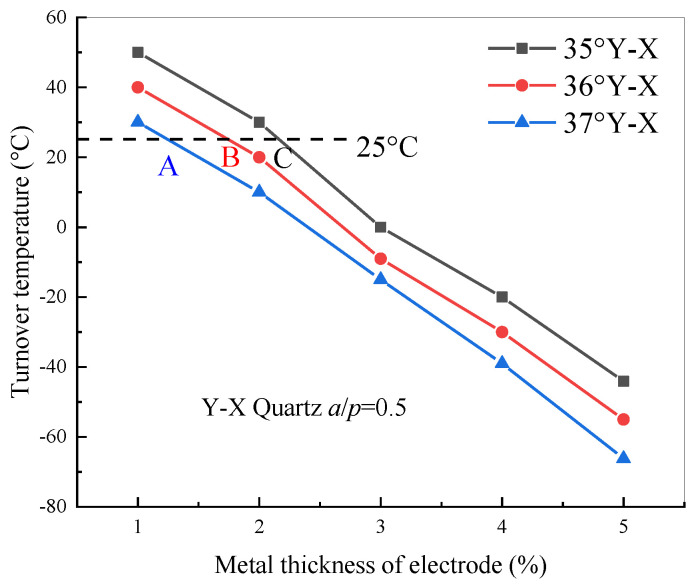
Turnover temperature of quartz with different cut angles and IDT with the electrode thickness.

**Figure 6 sensors-20-04237-f006:**
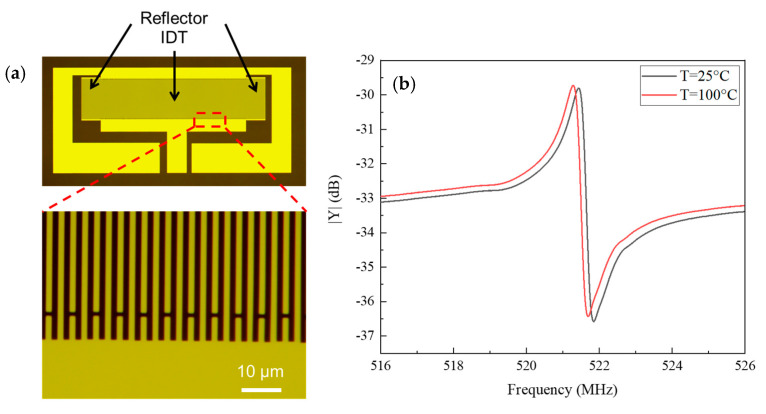
(**a**) Optical image and (**b**) the measured frequency response of a one-port SAW resonator based on quartz.

**Figure 7 sensors-20-04237-f007:**
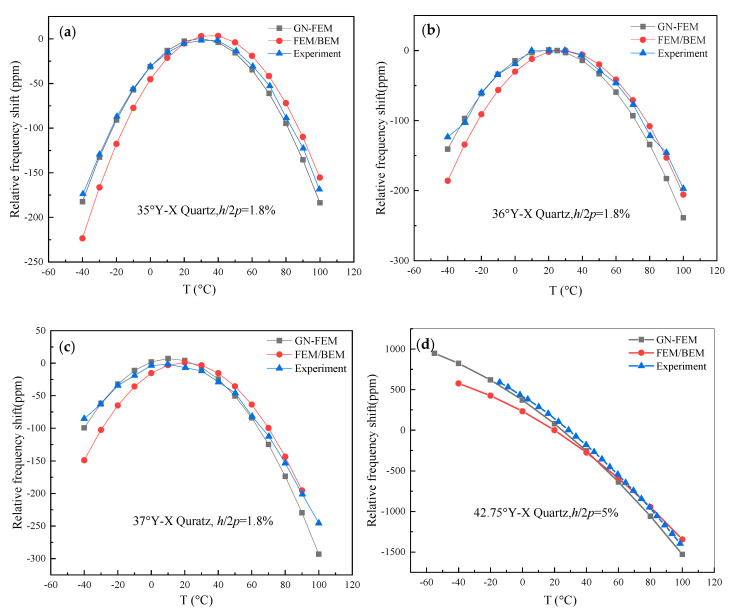
Comparison of the theoretical result and the experimental relative frequency shift of the quartz resonator. (**a**) 35.0° Y-X quartz; (**b**) 36.0° Y-X quartz; (**c**) 37.0° Y-X quartz; (**d**) 42.75° Y-X quartz.

**Figure 8 sensors-20-04237-f008:**
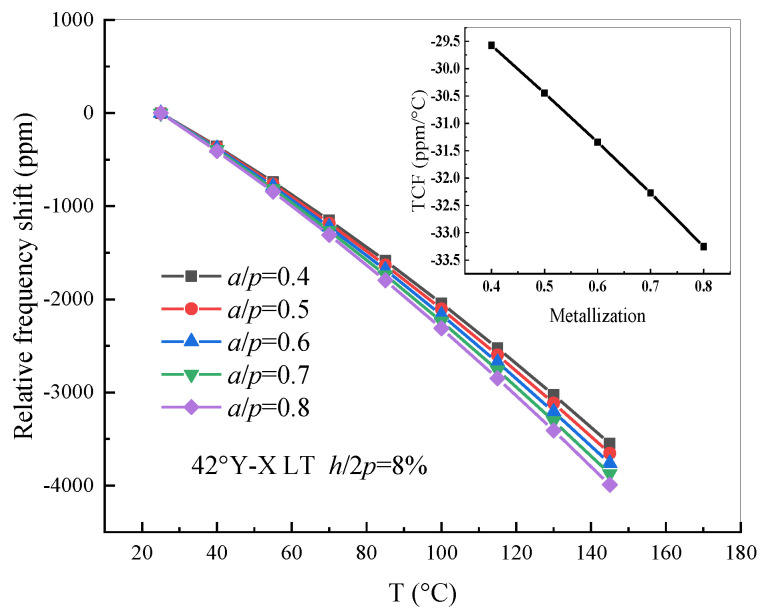
The relative frequency shift of 42° Y-X LT with different metallization ratios.

**Figure 9 sensors-20-04237-f009:**
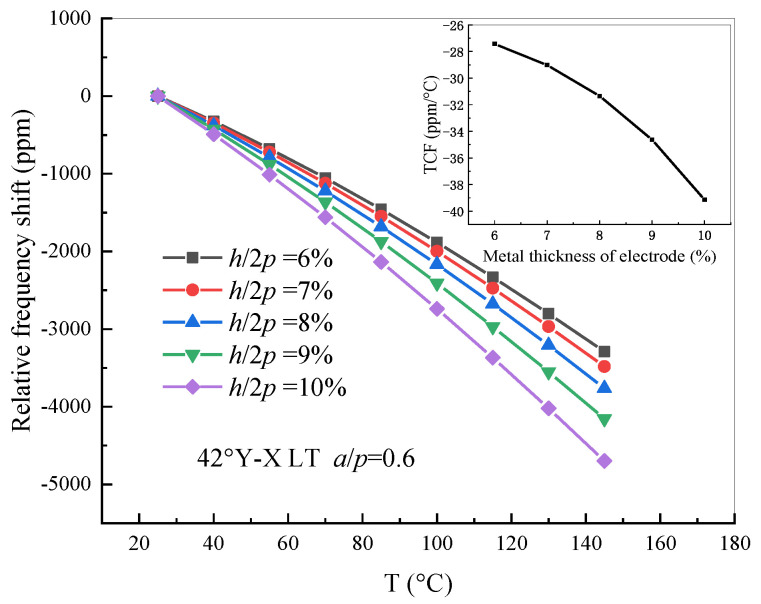
The relative frequency shift of 42° Y-X LT with different electrode thicknesses.

**Figure 10 sensors-20-04237-f010:**
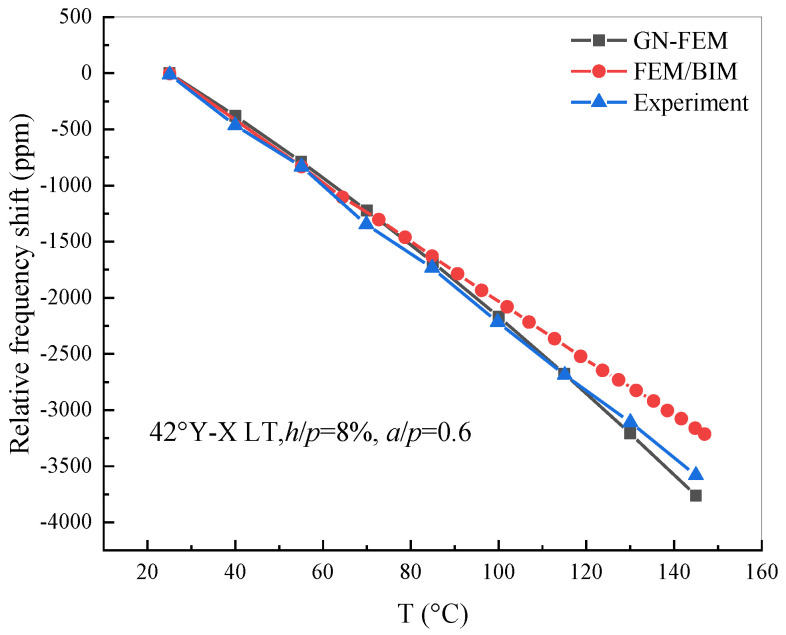
Comparison of the theoretical results and experimental relative resonance frequency shift of 42° Y-X LT.

**Table 1 sensors-20-04237-t001:** Detailed information on the above experimental sample.

Parameter	Value	Description
*p*	3 μm	Grating pitch
*a*/*p*	0.5	Metallization ratio
*h*/(2*p*)	1.8%	Metal thickness of electrodes
*n*_IDT	150	Number of IDT
*n*_GR	40	Number of grating reflectors
*W*	80 p	Aperture
Cut	42.75°	Cut angle of Y-X quartz

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
