# Peer review of "Geometric Nonlinear Model for Prediction of Frequency–Temperature Behavior of SAW Devices for Nanosensor Applications"

_sensors, 2020, doi:10.3390/s20154237_

Round 1

Reviewer 1 Report

Overall this is a good modeling paper to address the applications of quartz based SAW device. There's some minor things to revise. 

(1) Proof-read one more time. There's still a couple of English writing issues. (2) Label the dimensions on Figure 1.

(3) Address the terms such as a, h, p in the main text in the article. In the theoretical analysis, it's not addressed clearly how these parameters will impact the performance. Starting from Figure 2, all those parameters start showing in the results. Therefore, it will be better to include theoretical equations to demonstrate the impact. 

(4) Include the frequency information related to the cut angle for Quartz and LT.

(5) Needs to address how temperature range authors studied is related to harsh environment sensor.

Reviewer 2 Report

English writing requires extensive editing especially section 1 (introduction). 

It is not clear how this work is different from previously published research. Introduction section reviews previously published work but does not clearly distinguish contribution of this work. 

Figure 2 a,b,c, & d should be explained in caption. It is not clear why 42.75 Y-X Quartz cut results are not included in figures 3 & 4. 

It is not clear from Figure 6 that GN-FEM method provides better numerical solution than FEM/BEM method. For example, at high temperatures FEM/BEM method are close to experimental values. Best way to compare is use plot relative errors between numerical and experimental values. 

Reviewer 3 Report

The paper introduces a numerical tool based on FEM and Lagrangian equations for the thermal analysis of SAW devices on quartz and lithium tantalate. The accuracy is verified with experimental work. The 3D based FEM model that includes geometric nonlinear effects gives accurate results when compared to experiment.

The following issues need to be addressed, before the manuscript can be further considered for publication:

1. First, please check and correct the entire text, unfortunately there are plenty of grammar mistakes and typos which make the article very difficult to read. I give here some examples:
- "respectivey"(p1); "Miniaturization" should be written without capital M (p.1); "because Forsensors design" should be "For sensors design"(p.2); "proposeda general approach" (p.2); "In this study, the purpose of this paper is to develop" please choose either "in this study" or "the purpose of this paper"; "are in well good agreement" (p.2); "trapezoid shape of electrode combined
practical processing technology is proposed" (p2); "The remainder of this paper" (p2); "which lead to the mesh node of FEM model moves" (p3), etc

2. In Fig. 1 the trapezoid form of the IDT is not evidenced.

3. The authors state in the title, abstract and introduction that the paper investigates the thermal sensitivity for surface acoustic
wave devices for nanosensor applications. One would expect sub-micron IDTs if the target is the nanosensor. But the wavelength is 7 um. Also, I would say that SAW nanosensors on quartz are difficult to be realized due to technological constraints imposed by lithography. Please explain what are the targeted dimensions and how do you see the realization of a nanosensor on quartz or lithium tantalate.

4. As you represent in Fig. 2 the relative frequency shift (ppm) as a function of Temp, I would expect a frequency domain analysis. If this is the case, how did you compute the frequency response, having only one IDT? What are the boundary conditions imposed in the terminal? What is the resonance frequency extracted from FEM? I would consider these information more important than the number of mesh elements.

5. In section 3.2 please provide an optical photo of the SAW resonator that was manufactured on quartz and also S-parameter measurements to emphasize the resonance.

Round 2

Reviewer 2 Report

The paper still has a number of English writing issues for example

  • "The frequency–temperature characteristic of surface acoustic waves excited on piezoelectric substrates has been studied using many methods". should replace word "many" with "different"
  • This paragraph needs major revision: "The methods reported in the above previous works offer solution for prediction of frequency–temperature characteristic, but they ignored the nonlinear effect when thermal expansion. Lately, a remarkable solution of a weak form of nonlinear FEM model for calculating the thermal sensitivity on arbitrary layered structures has been reported [29], which pointed out the nonlinear is not negligible".
  • The following sentence is very long, "Hence, the geometric nonlinear model with the combination of thermal stress and strain tensors is employed to build the thermomechanical equilibrium equation on the mesh node of the FEM model after the mesh is moved. Moreover, the
    thermal stress and strain tensors of the substrate and electrodes, as well as the material constants, including the elastic constant, coupling constant, dielectric permittivity constant, and density as a
    function of temperature, are adequately taken into consideration."
  • "Based on an analysis of Figure 4, the turnover temperature of quartz with different electrode thicknesses was is shown in Figure 5. It is shown that the turnover temperature is quickly decreased as the electrode thickness and cut angle of quartz increase. The results of a comparison of Figure 3 and Figure 5 showed that the variation of electrode thickness has a larger effect on the frequency–temperature characteristic than that of the metallization ratio".
  • "For demonstration purposes, experimental samples based on a piezoelectric quartz substrate exciting Rayleigh-type SAW using the rotated Y-cut angles of 35°, 36°, 37°, and 42.75° at an operating
    temperature ranging from -40 to +100 °C were tested."

        It's not clear what does "exciting Rayleigh-type SAW" mean here. 

Reviewer 3 Report

The authors have improved the paper essentially. Still there are issues that need to be addressed:

1. It is true that the photolithography technology is relatively mature and there are lots of SAW devices developed on layered structures (e.g.AlN or ZnO on high acoustic velocity substrates) with submicron IDT widths.

The reference [16] currently cited in the paper also claims that the SAW devices developed on bulk piezoelectric substrates such as LiNbO3 and LiTaO3 are limited to a resonance of 3GHz due to the fact that the miniaturization of IDTs is limited by the resolution of the optical
lithography process employed in SAW mass fabrication today.

The authors claim in the title that the SAW devices are for nanosensor applications. Even if the title is resonant, for the moment the devices presented in the paper are developed on bulk conventional piezoelectric substrates - quartz or LT thin-plates with a wavelength of 7um. So, I suggest the authors to change this to "microsensor applications". In this way the title will be more suggestive for the readers.

2. I would suggest the authors to add a short explanatory paragraph in Section 3.1, summarizing the response 4.

3. The highlighted phrase from Introduction "The methods reported in the above previous works offer solution for prediction of frequency–temperature characteristic, but they ignored the nonlinear effect when thermal expansion." sounds unfinished. Please correct it.
